# Identification of *NHLRC1* as a Novel AKT Activator from a Lung Cancer Epigenome-Wide Association Study (EWAS)

**DOI:** 10.3390/ijms231810699

**Published:** 2022-09-14

**Authors:** Christian Faltus, Angelika Lahnsteiner, Myrto Barrdahl, Yassen Assenov, Anika Hüsing, Olga Bogatyrova, Marina Laplana, Theron Johnson, Thomas Muley, Michael Meister, Arne Warth, Michael Thomas, Christoph Plass, Rudolf Kaaks, Angela Risch

**Affiliations:** 1Division of Cancer Epigenomics, DKFZ–German Cancer Research Center, 69120 Heidelberg, Germany; 2Division of Cancer (Epi-)Genetics, Department of Biosciences and Medical Biology, University of Salzburg, 5020 Salzburg, Austria; 3Cancer Cluster Salzburg, 5020 Salzburg, Austria; 4Division of Cancer Epidemiology, DKFZ-German Cancer Research Center, 69120 Heidelberg, Germany; 5Translational Lung Research Center Heidelberg (TLRC), Member of the German Center for Lung Research (DZL), 69120 Heidelberg, Germany; 6Departament de Ciències Mèdiques Bàsiques, Universitat de Lleida, 25198 Lleida, Spain; 7Thoraxklinik at University Hospital Heidelberg, University of Heidelberg, 69126 Heidelberg, Germany; 8Institute of Pathology, Heidelberg University Hospital, 69120 Heidelberg, Germany

**Keywords:** EWAS, DNA methylation, lung cancer, *NHLRC1*, PI3K pathway activation

## Abstract

Changes in DNA methylation identified by epigenome-wide association studies (EWAS) have been recently linked to increased lung cancer risk. However, the cellular effects of these differentially methylated positions (DMPs) are often unclear. Therefore, we investigated top differentially methylated positions identified from an EWAS study. This included a putative regulatory region of *NHLRC1*. Hypomethylation of this gene was recently linked with decreased survival rates in lung cancer patients. HumanMethylation450 BeadChip array (450K) analysis was performed on 66 lung cancer case-control pairs from the European Prospective Investigation into Cancer and Nutrition Heidelberg lung cancer EWAS (EPIC HD) cohort. DMPs identified in these pre-diagnostic blood samples were then investigated for differential DNA methylation in lung tumor versus adjacent normal lung tissue from The Cancer Genome Atlas (TCGA) and replicated in two independent lung tumor versus adjacent normal tissue replication sets with MassARRAY. The EPIC HD top hypermethylated DMP cg06646708 was found to be a hypomethylated region in multiple data sets of lung tumor versus adjacent normal tissue. Hypomethylation within this region caused increased mRNA transcription of the closest gene *NHLRC1* in lung tumors. In functional assays, we demonstrate attenuated proliferation, viability, migration, and invasion upon *NHLRC1* knock-down in lung cancer cells. Furthermore, diminished AKT phosphorylation at serine 473 causing expression of pro-apoptotic AKT-repressed genes was detected in these knock-down experiments. In conclusion, this study demonstrates the powerful potential for discovery of novel functional mechanisms in oncogenesis based on EWAS DNA methylation data. *NHLRC1* holds promise as a new prognostic biomarker for lung cancer survival and prognosis, as well as a target for novel treatment strategies in lung cancer patients.

## 1. Background

Lung cancer represents the leading cause of cancer-related deaths. Despite the improvements in therapies, it still accounts for about one fifth of all cancer deaths [1,2]. According to the World Health Organization (WHO), the mortality rate of lung cancer was 83% in 2018 [3]. This observation probably stems from the fact that most lung tumors are diagnosed at advanced stages III and IV [4]. Most patients suffering from lung cancer are former or current smokers. However, the lifetime risk for developing lung cancer varies between studies from 6.7% in males and 4.1% in females [5] to 8.8% in males and 6.5% in females [6]. This reflects inter-individual differences in lung cancer susceptibility, which are not only underlined by results from genome-wide association studies (GWAS) [7], but also by changes in DNA methylation upon smoking [8,9,10,11].

Epigenome-wide association studies (EWAS) have been conducted for a wide range of diseases such as cancer, which are, at least in part, linked to the individual lifestyle [12,13]. EWAS have the common aim to identify epigenetic variants by means of blood DNA methylation changes on an epigenome-wide level in pre-diagnostic samples to estimate their association with disease risk. Recently, a multicenter lung cancer EWAS identified CpGs sensitive to smoke exposure which were hypomethylated in pre-diagnostic blood from lung cancer patients. Stringent analyses point out that the observed hypomethylation may explain the effect of tobacco exposure on lung cancer risk [9,10,11]. 

The Heidelberg sub-cohort of the European Prospective Investigation into Cancer and Nutrition (EPIC HD) provided one of the validation sets applied in two previous lung cancer EWAS [9,14], which found the strongest associations of cg05575921 and cg03636183 hypomethylation with lung cancer risk. In EPIC HD, we obtained HumanMethylation450 BeadChip array (450K) data of circulating lymphocyte DNA from 66 healthy pre-diagnostic lung cancer cases and an equal number of matched control subjects. In this discovery cohort, we aimed to identify DNA methylation alterations associated with lung cancer. The top differentially methylated positions (DMPs) and differentially methylated regions (DMRs) in lymphocyte DNA identified from EPIC HD were tested for differential methylation in lung tumor versus adjacent normal lung tissue using existing data sets from The Cancer Genome Atlas (TCGA) as well as two independent replication sets. We primarily focused on those DMPs and DMRs located in regulatory regions important for gene expression (i.e., close to annotated transcription start sites). To address functional roles of these DMRs in lung tumorigenesis we performed in depth in vitro characterization in lung cancer cell lines and report here on a novel gene, *NHLRC1,* involved in AKT activation. 

## 2. Results

### 2.1. Differentially Methylated CpGs from EPIC HD EWAS

In order to identify differentially methylated sites in lung cancer patients versus controls, 66 pre-diagnostic blood samples from smoking EPIC Heidelberg participants (Table 1), who developed lung cancer at a later point of time, were screened together with 66 matched controls with Illumina HumanMethylation450 BeadChip arrays (450K).

EPIC HD differential methylation was calculated in 63 sample pairs as intra-pair differences and resulted in the identification of 1,106 significantly differentially methylated positions (DMPs). The autosomal genome-wide *p*-values of intra-pair methylation differences for every CpG probe on the 450K array are displayed in Figure 1A. The vast majority of those CpGs were hypermethylated and a particularly high fraction of the hypermethylated DMPs was found in gene regulatory regions close to transcription start sites (TSS, Figure 1B). By contrast, gene desert regions showed more hypomethylated sites. We ranked EPIC HD DMPs by mean differential methylation across all sample pairs. The six top hypo- and hypermethylated DMPs listed in Table 2 were further considered for analysis comparing lung tumor tissue with normal adjacent tissue from TCGA data.

### 2.2. Top Hypermethylated EPIC HD EWAS DMPs Are Located within DMRs in Lung Tumor versus Adjacent Normal Tissue

Analysis of TCGA 450K data from lung squamous cell carcinoma (SCC, LUSC) and lung adenocarcinoma (ADC, LUAD) of the top-ranked EPIC HD DMPs revealed that these candidate CpGs were also differentially methylated in lung tumor versus normal tissue. Strikingly, the top two hypomethylated (cg22586603, cg13291208) and the top two hypermethylated (cg06646708, cg17242351) CpGs from EPIC HD were highly significantly differentially methylated in lung TCGA data sets. While differential methylation in regions located in the open sea can also impact cancer risk, e.g., by overlapping an enhancer or an insulator region, for this study we focused our analysis on those CpGs that were annotated as being proximal to a TSS and for which additional 450K probes existed within a region of 500 bp up- or downstream, in order to maximise chances of focusing on regions where differential methylation has an impact on gene expression. This limited our analyses to cg06646708 located 374 bp upstream of the annotated TSS of the ubiquitin E3 ligase NHL Repeat Containing 1 (*NHLRC1,*
Figure 2A) and to cg17242351, intragenic to the actin alpha cardiac muscle 1 (*ACTC1*, Appendix A). In addition, the latter overlaps an annotated transcript for the long non-coding RNA *LOC101928174.*

Detailed analysis of the *ACTC1* region showed that three CpGs (cg10580056, cg03844894 and cg05432213) adjacent to cg17242351 were hypermethylated in lung cancer versus normal tissue and in addition, *ACTC1* expression was reduced in lung tumor tissue from TCGA (Appendix A). cg06646708 is framed by two 450K probes (cg18068140 and cg18232313) within the *NHLRC1* upstream region, which were uniformly hypomethylated with elevated gene expression in lung tumor tissue from TCGA (lung adenocarcinoma (ADC): Figure 2B,C and squamous cell carcinoma (SCC): Appendix A. Hypomethylation of this DMR was also observed in TCGA bladder, liver and breast tumors (Appendix A). Notably, the 450K analysis of EPIC HD blood samples had revealed cg06646708 as hypermethylated, which is contrary to the TCGA data set. Given that this site was among the top ranked DMPs in the EPIC HD study, appears as hypervariable in blood samples as well as in tumors, and may be involved in regulating *NHLRC1* gene expression in lung tissue, we nevertheless considered it as worthy of further investigation.

To validate the *ACTC1* and *NHLRC1* lung tumor DMRs and to confirm their differential expression in lung tumors, we performed sequence-specific methylation-sensitive mass spectrometry (MassARRAY) and real-time PCR (rtPCR) in 94 lung squamous cell carcinoma (SCC) and in lung adenocarcinoma (ADC) versus paired adjacent normal lung tissue samples from current or former smokers of the lung tumor versus adjacent normal sample set (Table 3). By MassARRAY, we were able to quantitatively determine DNA methylation of a 399 bp region within the *ACTC1* gene surrounding cg17242351 (Appendix A) and a 256 bp region upstream of *NHLRC1* containing cg06646708 (Figure 2A). Thereby, we were able to confirm the hypermethylation of the *ACTC1* DMR in tumors and link it to decreased gene expression (Appendix A), as well as the *NHLRC1* DMR hypomethylation in line with the observed TCGA data (ADC: Figure 2D and SCC: Appendix A). Notably, *NHLRC1* expression was increased by 5.4-fold in ADC (Figure 2E) and 3.6-fold in SCC (Appendix A). By analysing an independent sample set of tumor versus adjacent normal (Replication Set II, Appendix A), we confirmed the *NHLRC1* DMR hypomethylation (Appendix A). The observed hypomethylation of the *NHLRC1* DMR in MassARRAY and increased gene expression is congruent with the TCGA data.

In conclusion, the EWAS results have pointed us to regions that may be functionally involved in deregulated gene expression in cancer. We hypothesize that the DMR containing cg06646708 upstream of *NHLRC1* regulates *NHLRC1* gene expression in lung tumor tissue. This was demonstrated by our MassARRAY and gene expression analysis in lung tumor versus adjacent normal lung tissue, as well as in TCGA data from different cancer tissues. Although the *ACTC1* DMR locus also shows substantial changes in DNA methylation and gene expression, we did not focus on this region further. Since this CpG overlaps with a long non-coding RNA, we cannot exclude a potential formation of a RNA:DNA hybrid (R-loop) influencing DNA methylation and gene expression via alternative pathways as shown elsewhere [15,16]. Thus, *ACTC1*, for which high expression is shown to indicate glioblastoma progression [17], remains a promising candidate to be investigated in a future study.

### 2.3. NHLRC1 Expression Is Epigenetically Regulated

To assess the regulatory potential of the identified DMR in *NHLRC1*, we analysed ChIP-seq data of activating and repressive histone marks from the Encyclopaedia of DNA Elements (ENCODE) project from the lung cancer cell line A549 and the normal lung fibroblast cell line NHLF [18]. We found the activating histone marks H3K4me1, H3K4me3 as well as H3K27ac enriched in the A549 lung cancer cell line compared to the normal lung fibroblast cell line NHLF, whereas the repressive mark H3K27me3 was absent in both cell types (Appendix A). This indicated that altered epigenetic regulation might be the cause for elevated *NHLRC1* transcription in tumor tissue.

To test the impact of DNA hypomethylation on *NHLRC1* transcription, we treated A549 and H1299 lung cancer cell lines and BEAS 2B normal lung cells with different concentrations of 5-Aza-2′-deoxycytidine (DAC), an approved epigenetic drug that leads to a global loss of DNA methylation through inhibition of DNA methyltransferases [19,20]. We used increasing DAC concentrations ranging from 10 nM to 1000 nM and determined the optimal DAC treatment efficacy for each cell line by measuring global DNA methylation of the long-interspersed elements (*LINE1*) by pyrosequencing (Figure 3A). H1299 and BEAS 2B showed the lowest *LINE1* methylation with 500 nM DAC and A549 with 100 nM DAC. Next, we analysed the methylation of the DMR upstream of *NHLRC1* (Figure 3B), which exhibited decreased methylation for DAC treated compared to the control cells concordant with the observed DNA demethylation of *LINE1*. The bronchial epithelial BEAS 2B cells showed DNA methylation resembling normal lung tissue methylation levels (Figure 3B, right panel). In contrast, the same region was less methylated in untreated A549 and H1299 lung cancer cells (Figure 3B, white bars of left and middle graphs). The *NHLRC1* baseline expression in lung tumor cells was approximately 2-fold higher compared to the expression level in normal bronchial BEAS 2B cells (Figure 3C). In line with this, we observed higher levels of *NHLRC1* expression induced upon demethylation in DAC-treated BEAS 2B cells compared to lung cancer cell lines (Figure 3D).

To confirm the regulatory activity of the *NHLRC1* upstream region we performed a sequence-specific DNA-methylation-dependent luciferase reporter assay comparing unmethylated and in vitro methylated *NHLRC1* upstream sequences. The *NHLRC1* upstream sequence showed a significant promoter activity, which was on average 11-fold (A549; Figure 3E) and 17-fold (H1299; Figure 3F) higher upon demethylation compared to the empty vector. Appendix A depicts the region tested with the Luciferase reporter assay, as well as the corresponding in vitro DNA methylation measurements with MassARRAY.

These results fit with the observation that chromatin remodeling via posttranslational histone modifications and transcription factor recruitment for the initiation of gene transcription occurs upstream of transcription start sites [21,22].

### 2.4. NHLRC1 Stimulates Proliferation, Viability, Migration and Invasion

Since *NHLRC1* was overexpressed in tumor tissue, we aimed to characterize *NHLRC1* functions in more detail. To do this, we performed RNAi mediated knock-down (KD) of *NHLRC1* in A549 and H1299 cells with highly specific siRNA pools [23]. *NHLRC1* siRNA KD was confirmed by rtPCR (Figure 4A,B). We measured numbers of proliferating and viable cells 96 h post-transfection. *NHLRC1* KD cells proliferated less (Figure 4C,D) and were less viable compared to control-treated cells (Figure 4E,F). In addition, *NHLRC1* KD attenuated transwell lung cancer cell migration (Figure 4G,H) and basement membrane invasion (Figure 4I,J). In brief, the overexpression of *NHLRC1* in tumor cells is associated with higher proliferation rates with an increased ability for migration and invasion, which is characteristic for cancer cells.

### 2.5. NHLRC1 Regulates AKT Activation and Modulates Expression of AKT Regulated Genes in Lung Cancer Cells

A striking observation in our RNAi experiments was the downregulation of phosphorylated AKT at serine 473 (pAKT_Ser473_) in A549 and H1299 cells upon *NHLRC1* KD (Figure 5A,D). The ratio of total AKT to pAKT_Ser473_ was approximately 1:1 in both control-treated lung cancer cell lines (Figure 5B,E), but in siRNA treated cells we found this ratio shifted towards 1:2 and 1:3, respectively. This indicates that *NHLRC1* loss alone was sufficient to attenuate oncogenic PI3K-AKT-mTORC2 signalling. To rule out an off-target effect of the *NHLRC1* siRNA pool, we transiently overexpressed *NHLRC1* or a *NHLRC1*-C26S catalytic mutant reported in [24] in H1299 cells. Notably, we observed a 2-fold upregulation of pAKT_Ser473_ only for cells overexpressing the wild-type *NHLRC1* but not for the C26S mutant (Figure 5G,H). This was still dependent on upstream PI3K-signalling since we found attenuated pAKT_Ser473_ by PI3K inhibitor treatment in H1299 cells (Figure 5G).

Previously, it has been shown that pAKT_Ser473_ translocates to the nucleus and phosphorylates FOXO transcription factors causing the export of FOXO from the nucleus [25]. Hence, active PI3K-mTOR signalling is crucial for cancer cell proliferation, since FOXO is prevented from regulating the expression of its target genes [26] as reviewed in [27]. Examples for FOXO-regulated genes are the apoptosis regulator tumor necrosis factor-related apoptosis-inducing ligand (*TRAIL* [28])*, BCL2 like 11* (*BIM* [29]), p53 upregulated modulator of apoptosis (*PUMA*, also known as *BCL2* binding component 3 [30]), cell cycle promoting genes CyclinD1 and CyclinD2 ([31,32]), RB transcriptional corepressor like 2 (*RBL2* [33]), cyclin dependent kinase inhibitor 1A (*CDKN1A* [34]), DNA repair enzyme growth arrest and DNA damage-inducible 45 (*GADD45* [35]).

Here, we investigated the effects of *NHLRC1* knock-down on gene expression levels of these *FOXO*-regulated genes on a protein basis. The analysis revealed that *TRAIL* was more than 100-fold upregulated in A549 *NHLRC1* KD cells (Figure 5C), but not expressed in H1299 cells (data not shown). Furthermore, *BIM* and *PUMA* were upregulated in A549 (Figure 5C) and H1299 cells (Figure 5F) upon *NHLRC1* KD. In addition, the FOXO-repressed genes CyclinD1 and CyclinD2 showed reduced expression in H1299 (Figure 5F) and to a lesser extent in A549 (Figure 5C). In contrast, *RBL2* and *CDKN1A* were only slightly increased in A549 (Figure 5C) and H1299 (Figure 5F). *GADD45* did not change upon *NHLRC1* KD in A549 (Figure 5C) and was only marginally increased in H1299 (Figure 5F). *CDKN1A* and *TRAIL* expression were not detected in H1299 cells (not shown). This suggests an activation of intra- and extracellular apoptosis signalling upon *NHLRC1* knock-down.

## 3. Discussion

Previously EWASs have identified DNA methylation markers associated with lung cancer risk [9,14]. Here, we investigated differentially methylated positions identified by Illumina HumanMethylation450 BeadChip arrays of prediagnostic blood samples from the EPIC Heidelberg cohort. The analysis of the top differentially methylated sites showed that they were also differentially methylated in lung tumors compared to adjacent normal tissue. Differential methylation at cg06646708 was of particular interest, since its hypomethylation was associated with overexpression of the closest gene, the ubiquitin E3 ligase *NHLRC1* not only in lung tumors, but TCGA data from different tumor sites. Recently *NHLRC1* methylation was linked with survival rates of SCC patients, with a decreased survival in patients upon lower methylation levels [36]. Comparing the consistent hypomethylation (and upregulation) in tumors to the hypermethylation (and downregulation) observed in the prediagnostic blood samples from cases vs. controls from 450K EPIC HD indicated that this is a tissue-specific hypervariable DNA methylation site. Hypervariability in DNA methylation in tumors at sites of methylation boundary shift in CpG shores has previously been associated with deregulated expression of cell cycle genes and as such these sites have been identified as being important regions on which to focus future epigenetic investigations [37]. *NHLRC1* was previously functionally characterized only in Lafora disease, a neurodegenerative type of myoclonus epilepsy [38,39]. Its cellular functions include the regulation of the mTOR pathway, microRNA processing body formation, glucose metabolism and autophagy [24,40,41]. However, these findings were linked to loss of function mutations in *NHLRC1* in the context of Lafora disease. Here, we demonstrated a novel mechanism in lung tumorigenesis resulting from epigenetic upregulation of *NHLRC* via DNA hypomethylation.

Several ubiquitin ligases have been linked to malignant alterations through deregulation of oncogenic cellular processes such as proliferation, apoptosis and cell cycle regulation [40]. *NHLRC1* was reported to contribute to p53 inactivation through nuclear to cytoplasmic translocation of homeodomain-interacting protein kinase-2 (*HIPK2*). Thus, elevated *NHLRC1* expression in lung tumor tissue may be a mechanism to inhibit TP53-pathway regulated induction of apoptosis in lung cancer [41]. The tripartite motif containing 32 (*TRIM32*), another RING-type ubiquitin E3 ligase and structurally highly similar to *NHLRC1* was shown to be deregulated in several malignancies leading to direct TP53 proteasomal degradation [42,43,44,45].

mTORC2 activation stimulates the central oncogenic downstream target AKT which triggers cell proliferation, survival, and chemotherapy resistance in lung and breast tumors [46]. Our results for *NHLRC1* knock-down in A549 and H1299 cells are in line with the consequences of pAKT_Ser473_ loss. The observed changes in gene expression of the pro-apoptotic genes *BIM*, *PUMA* and *TRAIL* in *NHLRC1* KD cells suggest that attenuated pAKT_Ser473_ leads to nuclear retention of FOXO3a. For instance, induction of *PUMA* causes intracellular apoptosis signalling in prostate cancer cells [47]. Similar effects were observed for lung cancer cells treated with cisplatin, a major first-line therapy for advanced non-small cell lung cancer [48,49]. Cisplatin interferes with PI3K signalling and stimulates the reactivation of FOXO3a, emphasizing the therapeutic importance of this mitotic cell signalling pathway. In summary, epigenetic deregulation of the DMR upstream of *NHLRC1* leads to its upregulation and, in turn, stimulates pAKT_Ser473_. This highlights the diverse routes for PI3K pathway activation apart from genomic mutations. Hence, the ubiquitin system represents a promising target for further investigation. In line with the work of Li et al., *NHLRC1* could be a promising new prognostic biomarker for lung cancer survival and prognosis, as well as a target for new treatment strategies in lung cancer patients [36].

## 4. Methods

### 4.1. Study Population

The epigenome-wide HumanMethylation450 BeadChip array (450K; Illumina, San Diego, CA, USA) was used for DNA methylation profiling of 66 healthy pre-diagnostic blood samples from incident lung cancer cases including adenocarcinoma (ADC), squamous cell carcinoma (SCC), small cell lung cancers and uncharacterized lung cancers and individually matched control blood samples from the Heidelberg component of the EPIC study (EPIC HD) conducted in accordance with the Declaration of Helsinki. Informed consent was obtained from all study subjects. The ethics committee of the Medical Faculty of the University of Heidelberg approved the use of these EPIC samples in this nested sub-study (S-627/2013) within EPIC (GEK; 13/94). Detailed information on EPIC HD is given elsewhere [9]. The present study was based on 211 incident lung cancer cases identified by July 2015. Never-smokers, study participants with any diagnosed neoplastic disease, as well as those with a lung cancer diagnosis less than one year after blood draw were excluded, in order to minimize tumor-specific changes in peripheral blood methylation. After accounting for these exclusion criteria, the 66 samples from lung cancer cases with the shortest time to lung cancer diagnosis were selected. Controls were individually matched to cases based on sex, age (±5 years), smoking status (current and former) and pack years of smoking (py; ±3py). The detailed sample set statistics are shown in Table 1 and included 25 ADC, 15 SCC, 19 small cell lung cancer and 4 uncharacterized lung cancers. 

Details on the lung tumor versus adjacent normal lung tissue replication set I and II compositions are given in Table 3 (Replication Set I) and Appendix A (Replication Set II). 

### 4.2. DNA Isolation

EPIC HD laboratory procedures were carried out at the DKFZ and LGC Bioscience. Buffy coat DNA was isolated at LGC Bioscience using the company’s standardized protocols and returned to DKFZ for the DNA methylation screen.

DNA of the tumor versus normal replication set was provided by Lung Biobank Heidelberg, a member of the accredited Tissue Bank of the National Center for Tumor Diseases (NCT) Heidelberg, the BioMaterialBank Heidelberg (BMBH) and the Biobank platform of the German Center for Lung Research (DZL). All participants gave their informed consent. The study was approved by the ethical committee of the Medical Faculty of the University of Heidelberg (Nr. 270/2001) and conducted in accordance with the Declaration of Helsinki. DNA was isolated with an AllPrep DNA/RNA Mini Kit (Qiagen, Hilden, Germany). Only tumor tissues with ≥50% viable tumor cells were used.

### 4.3. HumanMethylation450K BeadChip Array-Based Analyses of EPIC HD Samples

HumanMethylation450K BeadChip arrays were conducted by the DKFZ core facility for Genomics and Proteomics according to manufacturer’s instructions. Data analysis was conducted with RStudio (version 0.98.1091) [50] using RnBeads (version 0.99.17) [51]. Quality control measures included removal of probes overlapping known SNPs, probes not analysed in all samples or probes in non-CpG context, normalization with beta quantile dilation method (BMIQ) [52], as well as gender inference based on sex chromosome signal intensities. A total of 63 sample pairs passed the stringent quality control criteria and entered the differential methylation analysis. Blood cell type composition of every sample was estimated by a bioinformatics algorithm developed by Houseman and colleagues [53]. Principal component analysis of the cell type estimates was performed. The first two principal components were used for adjustment of observed intra-pair methylation differences in linear regression models for every CpG. *p*-values were corrected for multiple testing using the Benjamini Hochberg (BH) method [54]. A *p*-value threshold of 0.05 was applied and resulting CpGs were ranked by mean differential methylation across all sample pairs. 

### 4.4. HumanMethylation450K DNA Methylation Data Analyses of TCGA Data for Tumor versus Adjacent Normal Tissue

All 450K data for primary tumor and adjacent normal samples for lung ADC (n_Tumor_ = 361, n_Normal_ = 43) and lung SCC (n_Tumor_ = 424, n_Normal_ = 33) available by July 2014 were downloaded from the TCGA data portal. Differential methylation analyses were performed using the package RnBeads (version 0.99.17) [51]. Tumor versus adjacent normal data analyses for tumor entities other than lung were conducted using TCGA Wanderer [55].

### 4.5. Sequence-Specific Mass Spectrometry DNA Methylation Analysis (MassARRAY, Agena Bioscience, Hamburg, Germany)

A total of 500 ng genomic DNA was bisulfite-treated with EZ DNA Methylation kit (Zymo Research, Irvine, CA, USA). 1 µL was PCR-amplified in 5 µL reactions (Appendix A) with Hot Star Taq polymerase kit (Qiagen). PCR products were treated with shrimp alkaline phosphatase (SAP) and 2 µL SAP-treated PCR product were then in vitro transcribed with T7 polymerase, RNase A-cleaved, de-salted and finally subjected to matrix-assisted laser desorption/ionization time-of-flight (MALDI-TOF) mass spectrometry [56]. Knowledge of the expected sequence and mass distinction of the different fragments generated from the PCR amplicon allowed quantification of DNA-methylation for CpG unit, which contained 1 to 3 cytosines. Raw results are displayed as ratios of methylated and unmethylated detected fragments and were analysed by EpiTyper 2.0 analysis software. 

### 4.6. DNA Methylation Analysis by Pyrosequencing

Genomic DNA was bisulfite-treated as described above. PCR and sequencing primers were designed using Pyromark^®^ assay design software 2.0 (Qiagen) and sequenced on Pyromark^®^ Q24 (Qiagen) according to manufacturer’s instructions. The primer sequences are given in Appendix A. The data were analysed and exported with Pyromark^®^ Q24 Advanced software. 

### 4.7. RNA Isolation, DNase I Treatment, Reverse Transcription and Expression Analyses

RNA was isolated using the RNeasy mini kit (Qiagen) following the manufacturer’s instructions. RNA was DNase I -treated (Thermo Scientific, Waltham, MA, USA) and reverse transcribed with Oligo(dT)20 primers (Sigma Aldrich, St. Louis, MO, USA) with SuperScriptIII reverse transcriptase (Invitrogen, Carlsbad, CA, USA) according to the manufacturers’ instructions. Real-time quantitative PCR was performed with QuantiTect SYBR green (Qiagen) for tumor and adjacent normal samples on a LightCycler480 (Roche, Basel, Switzerland) and for lung cancer cell line RNA on a 7500 Real Time PCR machine (Applied Biosystems, Foster City, CA, USA). *NHLRC1* expression was normalized to the housekeeping genes *HPRT* and *GAPDH* (Appendix A). Primers for *BIM* and *PUMA* were previously published [57].

### 4.8. Routine Cell Culture and Treatments

A549, H1299 and BEAS 2B lung cell lines (ATCC, Manassas, VA, USA) were a gift from Dr. Ruprecht Kuner [58]. Lung cancer cells were cultured in Ham’s F12K medium (A549; Amimed—Bioconcept, Allschwil, Switzerland) and RPMI1640 medium (H1299; Thermo Scientific) with 10% FBS at 37 °C/5%CO_2_. Normal lung cells BEAS 2B were maintained in BEGM medium (BEAS 2B; LONZA) without FBS at 37 °C/5%CO_2_. Cells were routinely checked for mycoplasma contamination. 

#### 4.8.1. 5-Aza-2′-Deoxycytidine Cell Treatment

A549, H1299 and BEAS 2B were treated with 0.01-1 µM 5-Aza-2′-deoxycytidine (Decitabine—DAC; Sigma Aldrich) or DMSO (Thermo Scientific) for 72 h with cell culture media replacement every 24 h. Pyrosequencingof long interspersed elements such as *LINE1* confirmed treatment efficacy by DNA methylation loss.

#### 4.8.2. NHLRC1 Knock-Down and Overexpression

Knock-down of *NHLRC1* in A549 and H1299 cells was done with siRNA pools or negative control pools (siTools, Planegg, Germany) with RNAiMAX (Thermo Scientific) transfection reagent diluted in OptiMEM (Thermo Scientific) according to the manufacturers’ instructions.

*NHLRC1* coding region (Appendix A) was cloned into the pCRII-cGFP backbone expressing a GFP independently of the insert as transfection control. *NHLRC1* C26S catalytic mutant was generated with Quikchange XL II site-directed mutagenesis kit (Agilent, Santa Clara, CA, USA) according to the manufacturer’s instructions (Appendix A). Correct sequences were checked by Sanger sequencing (Eurofins MWG Operon, Ebersberg, Germany). We transiently transfected H1299 cells using Lipofectamine3000 transfection reagent (Thermo Scientific) according to the manufacturer’s instructions for 36 h. Cells were treated with 50 µM LY294002 PI3K inhibitor (Cell Signalling Technology, Danvers, Massachusetts, U.S.; CST) or DMSO (Thermo Scientific) control for 1 h prior to protein isolation.

### 4.9. Functional Assays

#### 4.9.1. Proliferation–Viability Assay

Proliferation assays using 4 µg/mL bisBenzimide (Hoechst 33342; Sigma Aldrich) were performed in vitro in A549 and H1299 cells treated with si*NHLRC1* and siRNA control pool and cell viability was determined by incubating cells with 0.1 µg/mL Calcein-AM (Sigma Aldrich) as previously published [59,60]. A live cell staining was performed at 37 °C/5% CO_2_ for 30 min followed by cell lysis (5 M NaCl, 10% TritonX in 1xPBS). Readouts were performed in a Tecan 200 plate reader (Grödig, Salzburg, Austria) at excitation/emission = 350 nm/460 nm for bisBenzimide and 485 nm/525 nm for Calcein in technical quadruplicates.

#### 4.9.2. Dual Luciferase Promoter Assay

Promoter luciferase assays were performed by cloning chr6:18122895-18123314 (Appendix A and Appendix A) into the pCpGfree-promoter-LUCIA plasmid (Invivogen, San Diego, CA, USA). In vitro *M.SssI* (Thermo Scientific) methylated or unmethylated pCpGfree-chr6:18122895-18123314 plasmids were co-transfected with pGL4-SV40-promoter (Promega, Madison, WI, USA) plasmid [59] into A549 and H1299 with the transfection reagent Lipofectamine 3000 (Thermo Scientific). Chemiluminescence was measured in a Tecan200pro plate reader (Tecan) in technical quadruplicates. Renilla readouts were normalized to Firefly luciferase and empty pCpGfree vector readouts.

#### 4.9.3. Transwell Migration–Invasion Assays

Migration–invasion assays were performed in si*NHLRC1*-treated A549 and H1299 cells. The 24 h post-starvation cells were seeded into 0.8 µM transwell-membrane chambers covered with (for invasion) or without (for migration) 1x basement membrane extract (Trevigen, Gaithersburg, MD, USA). The assay and the data analysis were performed adhering to the manufacturer’s instructions.

#### 4.9.4. Western Blot

Cells were lysed with RIPA buffer containing phosphatase inhibitors (Roche). Protein concentrations were estimated with BCA (BioRad, Hercules, CA, USA). Primary antibodies: 1:1000 (CST), secondary antibody anti-rabbit HRP: 1:5000 (CST). Detection: ECL reagent (GE Healthcare) and Gel Doc XR (BioRad). Images were analyzed and quantified by ImageJ [61] software. Raw blots are available in the Appendix A.

### 4.10. Statistical Analysis

*p* values for functional analysis including luciferase assay, *NHLRC1* siRNA knockdown and expression, invasion, and proliferation assays were calculated with a two-sided paired Student’s *t*-test with a confidence interval of 0.95 in RStudio [50].

## 5. Conclusions

In conclusion, this study has shown that there is a large potential for the discovery of novel functional mechanisms in oncogenesis based on EWAS DNA methylation data. The approach detailed here of analysing a top differentially methylated site from an EWAS study holds promise for the identification of further new mechanisms involved in cancer formation.

## Figures and Tables

**Figure 1 ijms-23-10699-f001:**
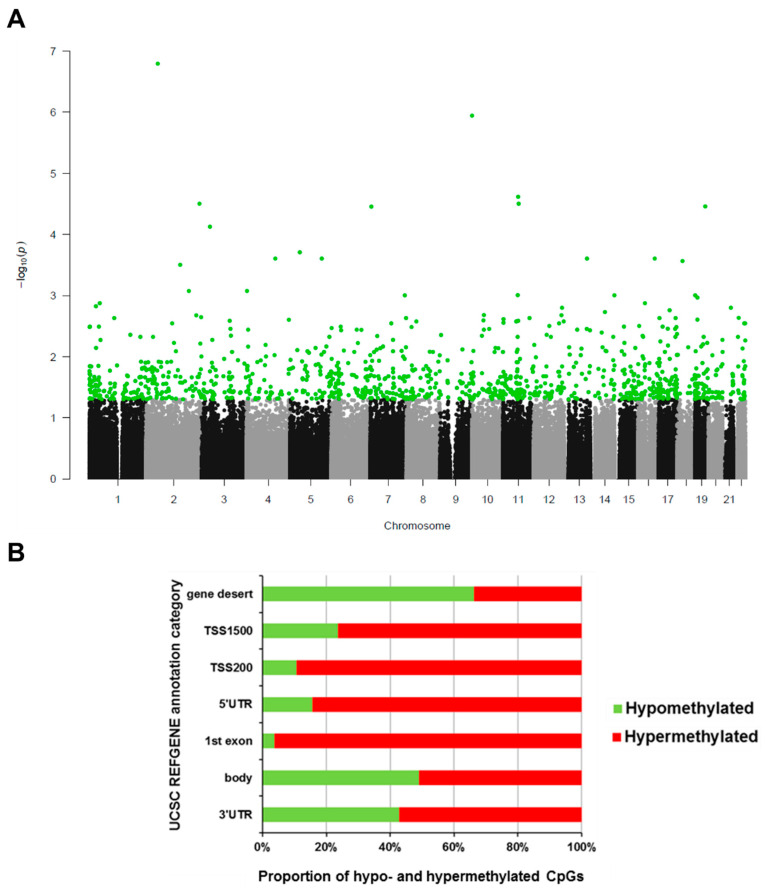
**DMPs in EPIC HD.** (**A**) Manhattan plot of genome-wide Benjamini Hochberg-corrected *p*-value distribution of 450K CpG methylation difference in EPIC samples. *p*-values of differential methylation in 63 pre-diagnostic lung cancer case versus healthy control blood sample pairs are plotted. Green dots indicate 1106 statistically significant probes below the BH corrected *p*-value threshold of 0.05. (**B**) Relative proportions of hypo- and hypermethylated CpGs by genomic context from the 1106 statistically significantly differentially methylated CpGs in EPIC HD. Annotation categories: gene deserts (none), up to 1500 bp (TSS 1500) or up to 200 bp (TSS 200) upstream of a transcription start site, 5′ untranslated region of a gene (5′UTR), first exon (1st exon), gene body (body) or the 3′ untranslated region (3′UTR).

**Figure 2 ijms-23-10699-f002:**
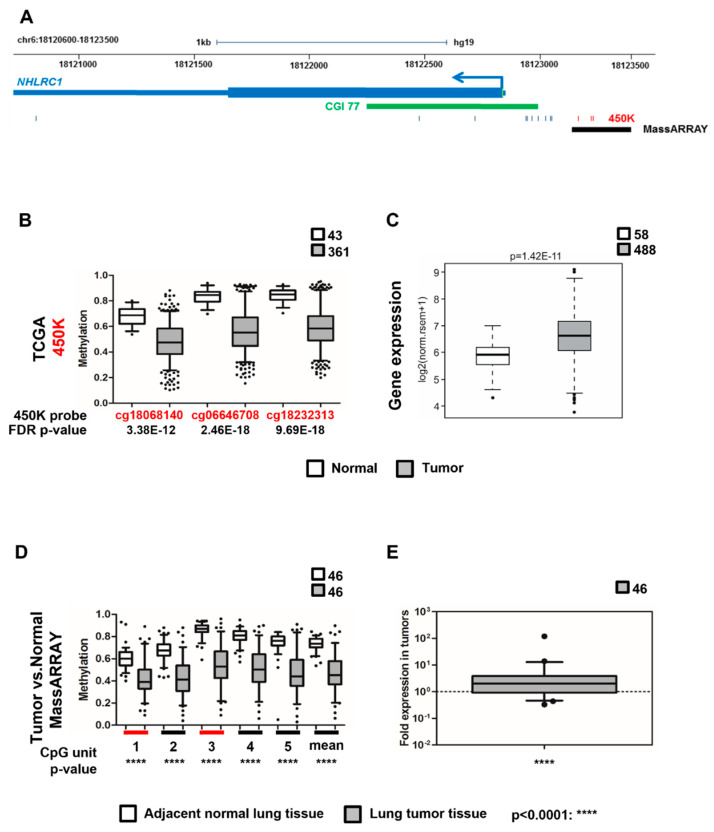
***NHLRC1* DNA methylation and mRNA expression in lung adenocarcinomas.** (**A**) Genomic location of the NHL repeat containing 1 (*NHLRC1*) gene. 256 bp region tested by MassARRAY and the 450K CpG probe positions are indicated. Red ticks depict differentially methylated 450K probes. (**B**) In silico TCGA tumor versus normal 450K data analysis for ADC for the candidate and 2 adjacent CpGs in the *NHLRC1* DMR. False discovery rate (FDR) corrected *p*-values are displayed. (**C**) *NHLRC1* gene expression of TCGA tumor versus normal data. (**D**) MassARRAY in 94 paired lung tumor versus 94 adjacent normal lung samples to determine DNA methylation of all CpGs in a 256 bp region displayed as CpG units. Red lines indicate those CpG units containing the CpGs which were originally identified as differentially methylated in EPIC and TCGA. Two-tailed *t*-test *p*-values. (**E**) *NHLRC1* mRNA expression analysis by real-time PCR in 92 samples of the lung tumor versus adjacent normal sample set. Whiskers indicate 5–95 percentile. Two-tailed *t*-test *p*-values, ****: *p* < 0.0001.

**Figure 3 ijms-23-10699-f003:**
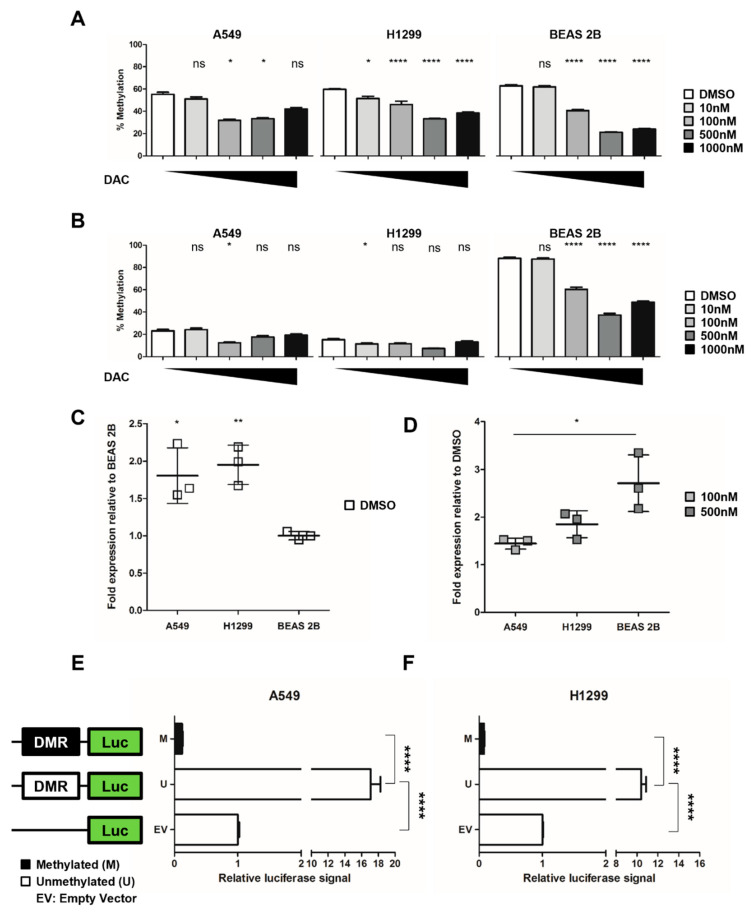
**DMR regulatory potential for *NHLRC1* mRNA transcription in lung cancer cell lines.** (**A**,**B**) Left and middle graphs: DNA methylation levels after 72 h treatments with 5-Aza-2’-Deoxycytidine (DAC) of A549 (left) and H1299 (middle) lung cancer cell lines. Right graphs: DNA methylation levels after 96 h DAC treatments of normal bronchial BEAS 2B cells. White bars = DMSO-treated cells. (**A**) LINE1 DNA methylation in DMSO-treated versus DAC-treated cells, determined by BS conversion followed by pyrosequencing. Bars represent the mean DNA methylation levels of 3 adjacent CpGs at the indicated DAC concentration. (**B**) *NHLRC1* DNA methylation pyrosequencing in DMSO versus DAC treated cells. Each bar represents the mean DNA methylation levels of cg06646708 and 4 adjacent CpGs at the indicated DAC concentration. (**C**) Baseline *NHLRC1* expression by real-time PCR in DMSO-treated cells relative to BEAS 2B (**D**) *NHLRC1* gene expression real-time PCR in DMSO versus 500 nM (H1299, BEAS 2B) and 100 nM (A549) DAC-treated cells. (**E**,**F**) Dual luciferase assays of *NHLRC1* upstream DMR region: (**E**) A549 cells, (**F**) H1299 cells. Fold luciferase activities induced by *NHLRC1* upstream DMR unmethylated (U) or in vitro methylated (M) sequence relative to empty vector (EV) are displayed. Two-tailed *t*-test *p*-values, *: *p* < 0.05, **: *p* < 0.01, ****: *p* < 0.0001, ns: not significant.

**Figure 4 ijms-23-10699-f004:**
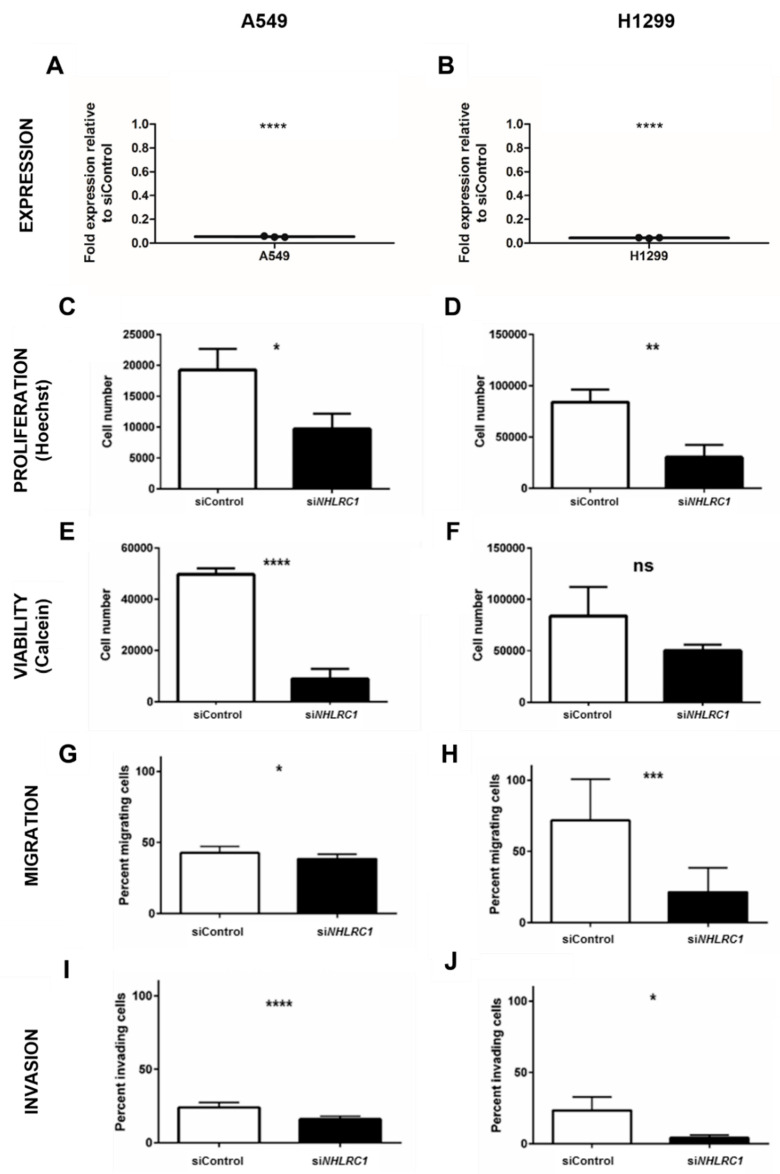
**Proliferation–viability assay and glucose uptake in *NHLRC1* knock-down lung cancer cells.** (**A**,**B**) Real-time PCR to confirm *NHLRC1* knock-down in A549 (**A**) and H1299 (**B**) relative to siRNA control-treated cells. *NHLRC1*: NHL repeat containing 1. siNHLRC1: NHLRC1 KD. siControl: Control siRNA treated cells. (**C**,**D**) Proliferation measured by Hoechst staining of A549 (**C**) and H1299 (**D**) in *NHLRC1* knock-down cells after 96 h after *NHLRC1* KD. (**E**,**F**) Viability of A549 (**E**) and H1299 (**F**) measured by Calcein-AM to Calcein turnover 96 h after *NHLRC1* KD. (**G**,**H**) Transwell migration–invasion assays of *NHLRC1* KD cells using FBS as a chemoattractant: 24 h migration of A549 cells through 0.8 µM pores (**G**) and 24 h migration of A549 cells through 0.8 µM pores (**H**). (**I**,**J**) 24 h basement membrane invasion of A549 (**I**) and H1299 (**J**) cells. Two-tailed *t*-test *p*-values, *: *p* < 0.05, **: *p* < 0.01, ***: *p* < 0.001, ****: *p* < 0.0001, ns: not significant.

**Figure 5 ijms-23-10699-f005:**
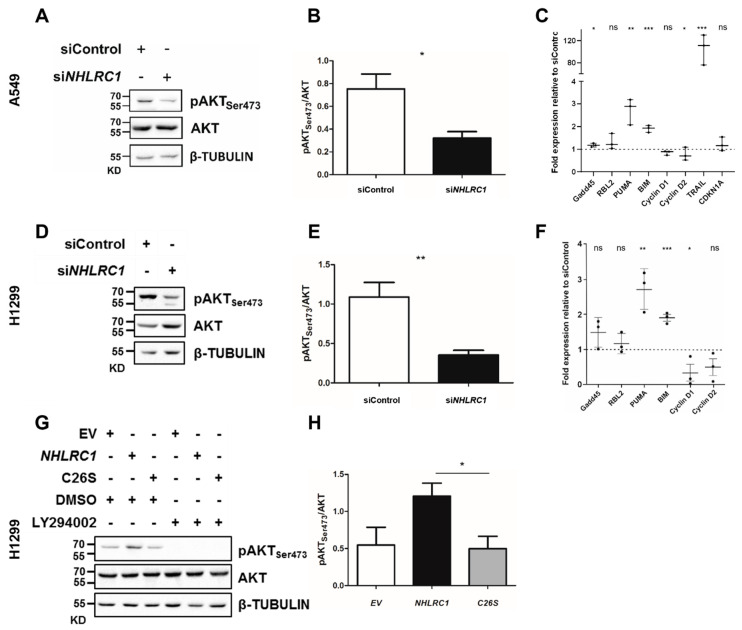
AKT activation and its downstream effects in lung cancer *NHLRC1* knock-down and overexpressing cells. (**A**,**D**) Representative Western blots probed for pAKT_Ser473_, AKT and beta-TUBULIN of lung cancer cell lines A549 (**A**) and H1299 (**D**) at 48 h after *NHLRC1* knock-down or control treatment. (**B**,**E**,**H**) Quantification of ratios from Western blot signals for pAKT_Ser473_ and AKT normalized to beta-TUBULIN. (**C**,**F**) Relative mRNA expression levels for FOXO-regulated genes in A549 (**C**) and in H1299 (**F**) *NHLRC1* knock-down cells versus control cells. (**G**) Representative Western blots probed for pAKT_Ser473_, AKT and beta-TUBULIN H1299 cells overexpressing at 36 h, compared to empty vector (EV) and *NHLRC1* C26S catalytic mutant (C26S) expressing cells. Cells were treated with 50 µM of the PI3K inhibitor LY294002 or with DMSO as control for 1 h. (**H**) Quantification of ratios from Western blot signals for pAKT_Ser473_ and AKT normalized to beta-TUBULIN. *NHLRC1*: *NHL* repeat containing 1, siControl: siRNA control-treated cells, si*NHLRC1*: *NHLRC1* siRNA pool-treated cells, KD: Kilo-Dalton, *GADD45*: growth arrest and DNA damage-inducible 45, *CDKN1A:* cyclin dependent kinase inhibitor 1A, *RBL2*: RB transcriptional corepressor like 2, *BIM: BCL2* like 11, *PUMA: BCL2* binding component 3; Raw blots are available as Appendix A and 2 for *NHLRC1* siRNA KD and Appendix A for *NHLRC1* overexpression experiments. Two-tailed *t*-test *p*-values, *: *p* < 0.05, **: *p* < 0.01, ***: *p* < 0.001, ns: not significant.

**Table 1 ijms-23-10699-t001:** Characteristics of the EPIC HD discovery blood sample set.

EPIC HD 450K Discovery Sample Set (Blood Samples)	
Total sample number cases and controls (n)	132
Mean age at baseline (years)	56 (range: 39–65)
Mean time from blood draw to diagnosis (years)	4.6 (range: 1.1–8.6)
Incident cases of lung cancer (n)	66
Males (n)	55 cases + 55 controls
Females (n)	11 cases + 11 controls
Current smokers (n)	48 cases + 48 controls
Former smokers (n)	16 cases + 16 controls
Mean pack years (n)	31.4 (range: 0.6–66.3)
Number of pairs considered in the analysis (n)	63 *

* 3 sample pairs did not meet quality control criteria.

**Table 2 ijms-23-10699-t002:** Top 1% (*n* = 12) of Ranked Hypo- and Hypermethylated CpGs from EPIC HD. BH: Benjamini Hochberg, hg19: human genome assembly 19.

**EPIC HD Top Hypomethylated CpGs**										
**Eank**	**450K Probe**	**Genomic Location (hg19)**	**Closest Coding Element**	**Distance to Closest TSS [bp]**	**EPIC HD Mean Methylation Difference [%]**	**EPIC HD BH *p*-Value**	**TCGA SCC Mean Methylation Difference [%]**	**TCGA SCC FDR *p*-Value**	**TCGA ADC Mean Methylation Difference [%]**	**TCGA ADC FDR *p*-Value**
1	cg22586603	chr8:129985596	*LINC00976*	7350	−8.9	2.7 × 10^−2^	−22.1	1.9 × 10^−12^	−11.9	2.0 × 10^−4^
2	cg13291208	chr8:70475388	*SULF1*	29,757	−6.6	3.6 × 10^−2^	−18.2	1.3 × 10^−6^	−23.4	1.2 × 10^−8^
3	cg06126421	chr6:30720080	*IER3*	7753	−6.3	2.4 × 10^−2^	−17.3	1.5 × 10^−8^	−4.4	3.7 × 10^−1^
4	cg13204432	chr2:51074939	*NRXN1*	184,735	−4.1	1.6 × 10^−7^	−9.1	4.5 × 10^−12^	−8.5	4.9 × 10^−6^
5	cg05951044	chr18:23597141	*SS18*	73,470	−4.1	2.7 × 10^−4^	−8.7	1.2 × 10^−6^	−3.6	4.2 × 10^−2^
6	cg04955914	chr2:220040571	*CNPPD1*	1144	−3.9	2.1 × 10^−3^	−6.0	5.0 × 10^−3^	−7.1	3.3 × 10^−2^
**EPIC HD Top Hypermethylated CpGs**										
**Rank**	**450K Probe**	**Genomic Location (hg19)**	**Closest Coding Element**	**Distance to Closest TSS [bp]**	**EPIC HD Mean Methylation Difference [%]**	**EPIC HD BH *p*-Value**	**TCGA SCC Mean Methylation Difference [%]**	**TCGA SCC FDR *p*-Value**	**TCGA ADC Mean Methylation Difference [%]**	**TCGA ADC FDR *p*-Value**
1	cg06646708	chr6:18123224	*NHLRC1*	374	6.7	3.2 × 10^−2^	−29.3	6.3 × 10^−25^	−23.9	2.5 × 10^−18^
2	cg17242351	chr15:35086890	*ACTC1*	1037	6.6	1.7 × 10^−3^	22.9	1.0 × 10^−8^	22.0	1.7 × 10^−10^
3	cg07714266	chr7:28075371	*JAZF1*	145,066	5.5	7.7 × 10^−3^	−14.4	4.5 × 10^−11^	−2.1	1.9 × 10^−1^
4	cg26484090	chr6:99960583	*USP45*	2669	5.4	2.0 × 10^-2^	−3.2	2.7 × 10^−3^	−1.2	3.2 × 10^−1^
5	cg00554165	chr5:172536348	*CREBRF*	52,993	5.2	4.7 × 10^−3^	4.6	1.7 × 10^−5^	0.2	5.8 × 10^−1^
6	cg14187813	chr2:97651611	*FAM178B*	690	4.9	3.6 × 10^−3^	−15.2	9.9 × 10^−9^	3.0	5.3 × 10^−1^

**Table 3 ijms-23-10699-t003:** Characteristics of the lung tumor versus adjacent normal lung tissue (Replication set I).

Lung Tumor versus Adjacent Normal Lung Tissue (Replication Set I)	
Total lung tumour-adjacent normal lung samples (n)	200
Adenocarcinoma (n)	50 (pairs)
Squamous cell carcinoma (n)	50 (pairs)
Mean age at diagnosis (years)	64 (range: 43–81)
Males (n)	68
Females (n)	32
Current smokers (n)	45
Former smokers (n)	54
Unknown smoking status (n)	1
Mean pack years (py)	46 (range: 1–150)
Number of patients considered in the analysis (n)	94 *

* 6 patients did not meet quality control criteria.

## Data Availability

Since participants of this study did not agree for their data to be shared publicly, supporting data is not available.

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
