# Peer review of "Identification of *NHLRC1* as a Novel AKT Activator from a Lung Cancer Epigenome-Wide Association Study (EWAS)"

_ijms, 2022, doi:10.3390/ijms231810699_

Round 1
Reviewer 1 Report
Alterations of DNA methylation during tumor progression have been recognized as important targets for the development of diagnostic and prognostic biomarkers for cancer. However, due to the heterogeneity of tumors, the analysis of loci differences based on epigenetics cannot provide a more in-depth explanation of the pathogenesis of the disease, making it difficult for these biomarkers to be validated and applied in clinical. In the initial work, EWAS microarray analysis was performed on clinical samples to validate differential methylation sites published in existing databases. The approach in this study which discovers novel functional mechanisms in oncogenesis based on EWAS DNA methylation data is impressive.
The work and results are interesting, while there are some major questions present as followings:
1. Due to the high-throughput screening of Chip arrays, it needs a detailed method description to explain how to eliminate false positive or false negative results in the Chip screening.
2. Analysis of candidate CpGs from Chip screening is a critical step in selecting molecules for subsequent validation experiments. In this study, the authors focused their analysis on those CpGs that were annotated as being proximal to a TSS and for which additional 450K probes existed within a region of 500bp up or downstream (lane 274). If it is a general analytical method, please add reference. If not, please explain the reason for this analysis.
3. In this article, only one molecule of NHLRC1 was screened for functional assays. Is it only one molecule has been screened? Why not select other potential molecules for functional verification?
4. For high expression of NHLRC1, in H1299 cells, an expression vector was transiently transfected into in H1299 cells (lane 199), it is recommended to use stable transfected cells for subsequent cell verification experiments.
5. Figure 5A: the result of beta-Tubulin on western-blot analysis seems not equal.
Author Response
We would like to thank all reviewers for carefully reading the manuscript and for their constructive comments. Please find attached our point-to-point reply.

Reviewer 2 Report
This is a beautiful work conducted by Christian Faltus and colleagues. In this project, the author found a novel factor, named NHLRC1, in regulating the lung cancer development process by screening of the blood samples from EPIC HD cohort. Through the epigenome-wide association study (EWAS), they identified a hypomethylated DMR region located in the promoter region of NHLRC1 gene in the lung cancer patient samples. They further explored the potential NHLRC1- related molecular pathway involved in the lung cancer development at cell line level, and found NHLRC1 downregulation can decrease the AKT phosphorylation thus leading to the attenuation of oncogenic PI3K-mTOR pathway activity and reduce the lung cancer cell migration and invasion. There findings could be helping to find new for new treatment strategies in 64 lung cancer patients. However, before its publication in IJMS, there are some improvements can be made, see my comments below:
1. In the method section(2.1), why the subjects with a lung cancer diagnosis less than one year after blood draw were excluded? Can the author explain further?
2. In Figure 3A and 3B, there are no statistic for these data. I didn’t see any significance or p-value here. Also, in Figure 5B,E, no statistics description or significance in the figure legend.
3. What’s the difference between A549 and H1299 cancer line? I saw they have big different response to the NHLRC1 knock-down, especially in the viability and migration assay( Figure 4)
Author Response

(The authors gave the same response as above.)

Reviewer 3 Report
In the present work, Faltus and colleagues address the role of NHLRC1 in lung cancer. They start from a whole-genome analysis, specifically an EWAS, and conclude with the characterization of the transcriptional regulation and of the biological function of NHLRC1 also highlighting its connection with the PI3K/AKT - FOXO axis.
The paper is well written, all the sections contain useful information in a detailed and concise format. All the performed experiments are consistent and well-designed.
There are only a few minor issues:
1- authors should state clearly if the samples used (both from EPIC cohort and from TCGA database) were of a specific subtype of lung cancer or if they considered all the subset
2- Figure 4G: carefully check the statistical significance. It looks strange that this pValue could be <0.05 while the difference depicted in panel F in defined as "NS"
3- Figure 5A: carefully check the normalization of this blot. Considering the amount of B-tubulin, it seems that also the level of AKT-tot decreases after siRNA treatment
4- Figure 5C: since TRAIL (and CDKN1A) are not expressed in H1299, the expression of this gene in A549 could be removed, for ease of interpretation. This would highlight the relevance of PUMA and BIM, which similarly increase in both cell lines.
Author Response

(The authors gave the same response as above.)

Round 2
Reviewer 1 Report
The authors have adequately addressed the questions raised in the 1st round of review, thus, it's recommended for publishing.